# Effect of Micro-Steps on Twinning and Interfacial Segregation in Mg-Ag Alloy

**DOI:** 10.3390/ma12081307

**Published:** 2019-04-22

**Authors:** Yi Liu, Xuefei Chen, Kang Wei, Lirong Xiao, Bin Chen, Haibo Long, Yandong Yu, Zhaohua Hu, Hao Zhou

**Affiliations:** 1Nano and Heterogeneous Material Center, School of Materials Science and Engineering, Nanjing University of Science and Technology, Nanjing 210094, China; Liuyi9280@163.com (Y.L.); weikang0106@163.com (K.W.); 2State Key Laboratory of Nonlinear Mechanics, Institute of Mechanics, Chinese Academy of Sciences, Beijing 100190, China; chenxuefei@lnm.imech.ac.cn; 3Institute of Microstructure and Property of Advanced Materials, Beijing University of Technology, Beijing 100124, China; fllhb0801@163.com; 4ThermoFisher Scientific, Shanghai Nanoport, Shanghai 201210, China; bin.chen@thermofisher.com; 5School of Materials Science and Engineering, Harbin University of Science and Technology, Harbin 150040, China; yandong_yu@163.com; 6Pangang Group Reserach Institute Co., Ltd., State Key Laboratory of Vanadium and Titanium Resources Comprehensive Utilization, Panzihua 617000, China; zhaohuahu2010@gmail.com

**Keywords:** magnesium alloy, twinning, HAADF-STEM, interfacial structure

## Abstract

Twinning structures and their interfacial segregation play a key role in strengthening of magnesium alloys. Micro-steps are frequently existed in the incoherent twin boundaries, while the effect of them on interface and interfacial segregation is still not clear. In this work, we performed an atomic-scale microstructure analysis using high-angle annular dark field scanning transmission electron microscopy (HAADF-STEM) to explore the effect of micro-steps on twin and its interfacial segregation in Mg-Ag alloy. Diffraction pattern of the incoherent {10$\bar{1}$1} twin shows that the misorientation has a slight tilt of 5° from its theoretical angle of 125° due to the accumulated effects of the micro-steps and their misfit dislocations in twin boundaries. Most of the micro-steps in {10$\bar{1}$1} twin boundary are in the height of 2$d_{\left(10\bar{1}1\right)}$ and 4$d_{\left(10\bar{1}1\right)}$, respectively, and both of them have two types according to whether there are dislocations on the micro-steps. The twin boundary is interrupted by many micro-steps, which leads to a step-line distributed interfacial segregation. Moreover, the Ag tends to segregate to dislocation cores, which results in the interruption of interfacial segregation at the micro-steps with dislocations.

## 1. Introduction

As the lightest metallic material, magnesium and its alloys are attracted by aeronautical, automotive, chemical, and medical industries [1,2,3,4,5]. In recent years, wrought magnesium alloys have been well developed due to their remarkable interfacial strengthening induced by high density of grain boundaries and twin boundaries [6,7,8]. Due to the hexagonal close packed (HCP) crystal structure, the number of easy-glide systems in Mg alloys are not enough to satisfy the uniform plastic deformation according to the Von Mises criteria. Thus, it is not possible for the strain to be totally accommodated by dislocation motion. Instead, it is done by twinning, which plays a key role in the deformation of magnesium alloys [9,10,11,12,13,14]. Moreover, it has also been reported that the interfacial segregation plays an important role in mechanical properties, especially in the thermal stability of the magnesium alloys. The segregated structure is not sensitive to the temperature and time of heat treatment, while it has four crystal structural factors, including the atom radius, the types of crystal, the distribution of stress field, and interfacial energy [15,16,17]. Nie et al. [15] found that the annealed twin boundaries were decorated by a periodic distribution of segregated solute atoms (Gd, Zn and Gd&Zn) due to the periodic stress distribution of twin boundaries. The specific segregation position of solute atoms is determined by the atomic radius of solute elements. Zhou et al. [16] reported a spinal-shaped segregation at {10$\bar{1}$2} twin boundary (TB) and high-angle lamellar grain boundary in Mg-Gd-Y-Ag-Zr alloy, which was induced by the high energy of the interfaces. Moreover, Xiao et al. [17] found an interfacial phase formed at high strain sites of the co-axial grain boundaries in Mg-Gd-Y-Ag alloy, which had a higher thermal stability than the precipitates in grain interior.

In general, most of the twin boundaries in Mg alloys are incoherent boundaries that have many micro-steps along the interfaces. Liu et al. [18] reported that the morphology of {10$\bar{1}$2} twin boundaries (TBs) in pure Mg were a zigzag-shape observed under a high-resolution transmission electron microscopy (HRTEM). Zhang et al. [19] found that the TBs in Co and Mg significantly deviated from the ideal {10$\bar{1}$2} twin planes. It has been revealed that the distinct deviation is result from the zigzag-shape steps formed by TBs and basal-prismatic interfaces [20,21,22]. In addition, dislocations are frequently observed at the zigzag steps. Serra et al. [23] claimed that the dislocations in the micro-steps with high energy were hard to glide. The above micro-steps and dislocations are very important structures in TBs, while their effects on interfacial segregation in Mg alloys are still unclear. In this work, the atomic scaled structures of micro-steps in TBs were investigated using a high-angle annular dark-field scanning transmission electron microscope (HAADF-STEM). We focused the attention on the effect of micro-steps on twinning misorientation and interfacial segregation in Mg-Ag alloy.

## 2. Materials and Methods

The Mg-2.57 wt.% Ag alloy was prepared by melting mixed high purity Mg and Ag (99.95%) in an electric-resistant furnace (The details of alloy preparation have been reported in [24]). The as-cast ingots were solution treated at 450 °C for 12 h, and then were quenched to room temperature in water. Samples with a dimension of 30 × 40 × 6 mm^3^ were used as the start material for hot rolling followed by cold rolling. Samples for hot rolling were heated at 500 °C for 10 min before each rolling pass and total rolling reduction of 35%. Cold rolling was performed at room temperature with 0.1 mm thickness reduction per pass to a total rolling reduction of 20%. Post annealing was carried out at 150 °C for 30 min to induce interfacial segregation.

Transmission electron microscopy (TEM) specimens were cut from the annealed rolling sheet and polished to a thickness of ~60 μm with sandpaper of 320, 800, and 2000 grits. The pre-thinning process was performed using a Dimple Grinder machine (Gatan Model 657). The final perforation was preformed using an ion milling instrument (PIPS 691, Gatan, Pleasanton, CA, USA) with low angle (<3.5°) and low energy ion beam (<3 keV). Atomic-resolution high-angle annular dark-field (HAADF) observation was carried out on an aberration-corrected scanning transmission electron microscope (STEM) (Titan G2 60-300, FEI, Hillsboro, OR, USA) operated at 300 kV. The zone axis of HAADF-STEM observation is parallel to [1$\bar{2}$10] of Mg matrix.

## 3. Results and Discussion

Figure 1a shows a low-magnification TEM image of the annealed Mg-2.57 wt.% Ag alloy samples after rolling. A “willow leaf” shaped structure is observed which has a length of ~2 μm and a width of ~300 nm. It is a typical morphology of deformation twin, which has a line-shaped interface with the matrix [25]. Figure 1b shows the selected area diffraction pattern (SADP) of the interface in [1$\bar{2}$10] zone axis. Two sets of co-axial diffraction patterns were obtained, which is similar to the one of {10$\bar{1}$1} TB [15,16]. However, the measured misorientation of the boundary is ~120°, while the ideal angle of {10$\bar{1}$1} twin should be 125°. Due to this 5° of deviation, the diffraction pattern of (10$\bar{1}$1) crystal planes were not overlapped completely, as marked in Figure 1b. Figure 1c shows the HAADF-STEM image of a TB, which has a zigzag shape with apparent solute element segregation according to the Z contrast of atoms. Many micro-steps are observed in the TB, which have an interval distance of ~10 nm, as marked by the white arrows in Figure 1c. Figure 1d is the atomic-scale HAADF-STEM image of interfacial segregation in a {10$\bar{1}$1} TB. It is found that the misorientation in such local area is ~124°, which is more closed to the ideal 125°. Interestingly, the misorientation deviation can be accumulated with the increase of micro-steps. Thus, the deviation angle of the boundary in the low magnified image is ~5° (Figure 1a,b). The slight tilt of TB is similar to the tilt boundary, which is induced by ordered accumulation of dislocations in grains [26,27].

According to the brightness of atomic columns in HAADF-STEM images (the Z contrast is proportional to the square of atomic number), the Ag-rich and Mg column were distributed alternately along the {10$\bar{1}$1} TBs as shown in Figure 1c,d. Nie et al. [15] have reported that the bigger Gd atoms tend to segregated at extension sites of TBs, while the smaller Zn atoms usually go to the compressed sites of TBs. The atomic radius of Ag is ~0.144 nm, which is less than that of Mg (~0.160 nm). It is found that Ag segregated at compressed sites of {10$\bar{1}$1} TB of Mg-Ag alloy.

The micro-steps along twin boundaries have different heights, and most of them are in the height of two or four {10$\bar{1}$1} atomic planes, which are represented by 2$d_{\left(10\bar{1}1\right)}$ and 4$d_{\left(10\bar{1}1\right)}$, respectively. Molecular dynamics simulation studies indicated that the most popular micro-steps in {10$\bar{1}$1} twins were in the height of 2$d_{\left(10\bar{1}1\right)}$, which had the lowest interfacial energy [28,29]. Figure 2a shows the HAADF-STEM image of {10$\bar{1}$1} TB from [11$\bar{2}$0] direction. The edge-on TB is marked by the blue dash line. It is clear that the height of micro-steps I and II are in the height of 2$d_{\left(10\bar{1}1\right)}$ and 4$d_{\left(10\bar{1}1\right)}$, respectively. Figure 2b is the lattice fringes image obtained from the inverse fast Fourier transform (FFT), which filters the (10$\bar{1}$1) diffraction spots in corresponding FFT pattern out, as shown in Figure 2b. Some misfit dislocations are observed at the micro-steps as marked by the red symbols of “**⊥**”. Note that both the micro-steps I and II have two types according to the whether and there are misfit dislocations at them. The zigzag morphology of twin boundary is induced by the micro-steps. The micro-steps were considered as twinning dislocations, which were characterized by the step height ***h*** and Burgers vector ***b*** [30]. Similar to the formation of low-angle tilt boundaries in single crystal [31,32], the twinning dislocations pile-up along a twin boundary, and cause a slight tilt from its theoretical misorientation. As shown in Figure 1b, the zone axes at both sides of the tilted {10$\bar{1}$1} TB are in the same direction. It is reasonable to imply that the tilt of TB is around the [11$\bar{2}$0] zone axis. Thus, the zigzag morphology of TB is resulted from the micro-steps along the TB, and the misorientation deviation is induced by the slight tilt effect of dislocations at micro-steps.

Figure 3a,b shows two atomic scaled HAADF-STEM images of {10$\bar{1}$1} TBs in [11$\bar{2}$0] zone axis, which have micro-steps I in the twin boundaries. The traces of twin plane (10$\bar{1}$1) and basal plane (0001) are marked by blue and white dash lines, respectively. Dislocation circuits are constructed at micro-steps based on these two reference planes to analyze whether there are dislocations on the micro-steps. As shown in Figure 3a, the Burgers circuit starts from point **S** at the twin plane, and goes through seven atomic columns to point **A** along (0001) basal plane to form the first side of the dislocation circuit. The second side of dislocation circuit is parallel to the trace of (10$\bar{1}$1) plane, which goes through 10 layers basal planes (starts from point **A** and ends by point **B**), as marked in Figure 3a. The rest of dislocation circuit is finished by points **C**, **D**, **E**, and **F**, while the point **F** does not overlap with the start point **S**. It indicates that the vector **FS** (marked by the red arrow in Figure 3a) is the Burgers vector of twinning dislocation (**b** = 1/3[10$\bar{1}$0]). Zhu et al. believed that the twinning dislocations at micro-steps were induced by the interaction of stacking faults and TBs [33]. Figure 3b shows another type of micro-step I which is free of dislocations. The dislocation circuit close perfectly, and the point **F** overlaps with the start point **S**. Interestingly, the interfacial segregation on these two micro-steps are very different. Figure 3a exhibits that the periodic segregation is interrupted at the micro-step with dislocation. Three atomic columns marked by the green arrows have no elemental segregation according to the Z contrast of HAADF-STEM image. In contrast, the Ag segregation was not affected by the micro-steps without dislocations, which have complete segregation structure in the same atomic columns, as shown in Figure 3b. The structure of these two different micro-steps are almost the same, except for the existence of twinning dislocation. Thus, the interfacial segregation is proposedly affected by the twinning dislocation.

Similar result was observed in the micro-steps II in height of 4$d_{\left(10\bar{1}1\right)}$. There are twinning dislocations at the micro-steps, which have the same Burgers vector of 1/3[10$\bar{1}$0], as shown in Figure 4a. The accumulated effect of dislocations with same Burgers vector result in a slight tilt of {10$\bar{1}$1} TB. Moreover, Figure 4a shows that the periodic interfacial segregation is interrupted at the position of twinning dislocations. Three typical atomic columns marked by the green arrows are free of element segregation. However, the interfacial segregation is complete at the micro-steps without twinning dislocation as shown in Figure 4b. According to the investigation on interfacial segregation of tilt boundaries, the solute atoms tended to segregation into the sites with high strain, such as dislocation cores, to reduce the total energy of the interface [33]. Thus, it is reasonable to explain that the interrupted interfacial segregation was affected by the attraction of the twinning dislocation cores. 

## 4. Conclusions

This study reveals interfacial structure of {10$\bar{1}$1} twin in annealed Mg-Ag alloy using HAADF-STEM. The key findings are summarized as follows:
Many micro-steps exit in incoherent {10$\bar{1}$1} TBs, which results in a zigzag morphology of TBs. The most popular types of micro-steps are in heights of 2$d_{\left(10\bar{1}1\right)}$ and 4$d_{\left(10\bar{1}1\right)}$. The misorientation of TBs shows a slight tilt around the [11$\bar{2}$0] axis, owing to the accumulated effect of twinning dislocations with the same Burgers vector of 1/3[10$\bar{1}$0].Interfacial segregation of {10$\bar{1}$1} TBs is affected by the micro-steps. First, the micro-steps affect the morphology the twin boundary to a zigzag shape. Consequently, the interfacial segregation also presents a step-like structure. Moreover, the periodic interfacial segregation is interrupted at the micro-steps which have twinning dislocations. The attraction of twin dislocation cores is proposed as the main reason for this effect, which may reduce the total energy of twin boundaries.

## Figures and Tables

**Figure 1 materials-12-01307-f001:**
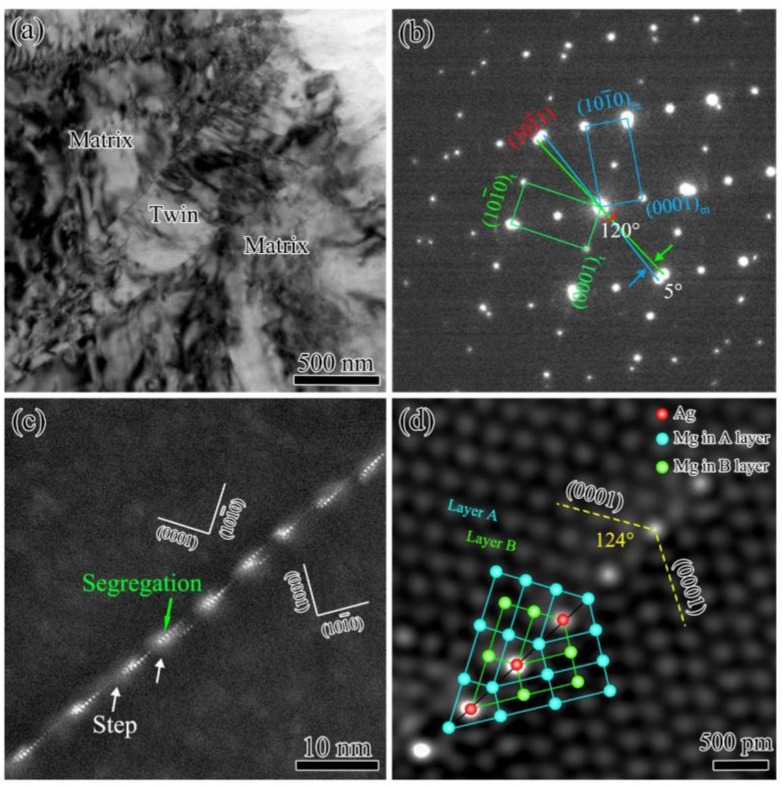
TEM images of {10$\bar{1}$1} twin in Mg-Ag alloy from [1$\bar{2}$10] direction: (**a**) A low magnified bright field TEM image; (**b**) a selected area diffraction pattern, (**c**) a high-resolution HAADF-STEM image, and (**d**) an atomic scaled HAADF-STEM image marked by an atomic model. The red spheres represent Ag atoms. Blue and green spheres represent Mg in A layer and B layer, respectively.

**Figure 2 materials-12-01307-f002:**
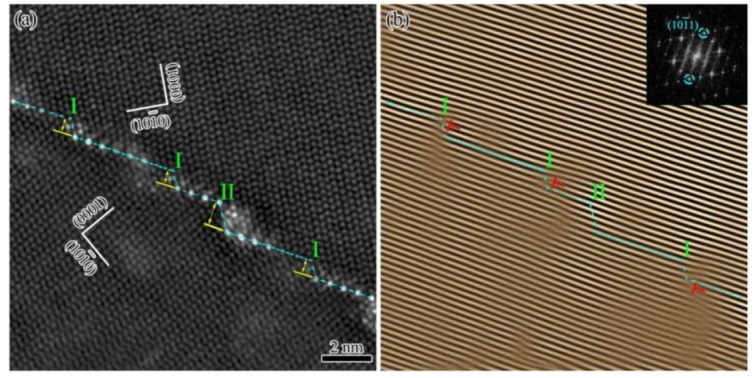
TEM images of {10$\bar{1}$1} twin boundary in [1$\bar{2}$10] zone axis: (**a**) an atomic-scale HAADF-STEM image with micro-steps; and (**b**) a lattice fringes images obtained by the inverse Fourier transform (IFFT) of (10$\bar{1}$1) diffraction spots of the HAADF-STEM image and the Fourier transform diffraction pattern.

**Figure 3 materials-12-01307-f003:**
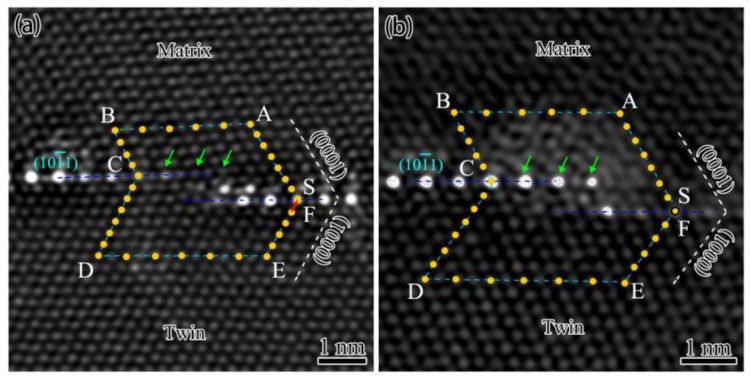
Atomic-scale HAADF-STEM images of micro-step I in [1$\bar{2}$10] zone axis marked with a Burgers circuit: (**a**) A micro-step with misfit dislocation, and (**b**) a micro-step free of dislocation.

**Figure 4 materials-12-01307-f004:**
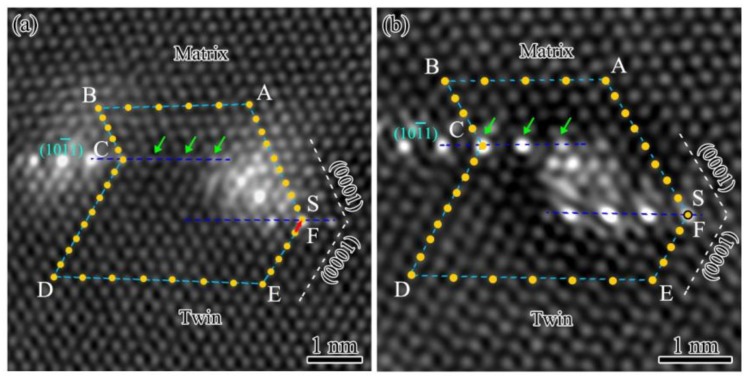
Atomic-scale HAADF-STEM images of micro-step II in [1$\bar{2}$10] zone axis marked with a Burgers circuit: (**a**) A micro-step with misfit dislocation; (**b**) a micro-step free of dislocation.

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
