# Peer review of "Effect of Micro-Steps on Twinning and Interfacial Segregation in Mg-Ag Alloy"

_materials, 2019, doi:10.3390/ma12081307_

Round 1
Reviewer 1 Report
The present manuscript investigate the effect of micro-stepping on the deformation behavior of the Mg-Ag alloy using TEM. The study is interesting and valuable. Can be accept as it is. However, the experimental details need to be improved, specifically the sample preparation for the TEM.
Author Response
Response 1: Thanks for the reviewer’s comments. We had added the experimental details on P2 and P3 in the revised manuscript, which has also been highlighted in yellow as follows.
Samples with a dimension of 30 × 40 × 6 mm3 were used as the start material for hot rolling followed by cold rolling. Samples for hot rolling were heated at 500 ℃ for 10 min before each rolling pass and total rolling reduction of 35%. Cold rolling was performed at room temperature with 0.1 mm thickness reduction per pass to a total rolling reduction of 20%.
The pre-thinning process was performed using a Dimple Grinder machine (Gatan Model 657). The final perforation was preformed using an ion milling instrument (Gatan PIPS 691) with low angle (<3.5°) and low energy ion beam (<3 keV).
Reviewer 2 Report
This paper revealed the interfacial structure of a specific twin in Mg-Ag alloy very well. The experimental results are very high quality, and the authors discussed the structure extensively. The reviewer read the paper interestingly. Please edit a few typing errors, for instance, 'defroamtion', 'it is clearly that', and 'Mang micro-steps' etc.
Author Response
Response 1: Thank you very much for your comments and we really apologized. It has been checked carefully throughout the manuscript and all the spelling errors have been revised and highlighted in colour.